# Template for Rapid Iterative Consensus of Experts (TRICE)

**DOI:** 10.3390/ijerph181910255

**Published:** 2021-09-29

**Authors:** Angel M. Chater, Gillian W. Shorter, Vivien Swanson, Atiya Kamal, Tracy Epton, Madelynne A. Arden, Jo Hart, Lucie M. T. Byrne-Davis, John Drury, Ellie Whittaker, Lesley J. M. Lewis, Emily McBride, Paul Chadwick, Daryl B. O’Connor, Christopher J. Armitage

**Affiliations:** 1Centre for Health, Wellbeing and Behaviour Change, University of Bedfordshire, Bedford MK41 9EA, UK; 2Centre for Behavioural Medicine, University College London, BMA House, Tavistock Square, London WC1H 9JP, UK; 3Centre for Improving Health Related Quality of Life, School of Psychology, Queen’s University Belfast, Northern Ireland BT7 1NN, UK; g.shorter@qub.ac.uk; 4Psychology Division, University of Stirling, Stirling FK9 4LA, UK; vivien.swanson@stir.ac.uk; 5NHS Education for Scotland, 2 Central Quay, 89, Hydepark Street, Glasgow G3 8BW, UK; 6School of Social Sciences, Department of Psychology, Birmingham City University, Birmingham B4 7BD, UK; Atiya.Kamal@bcu.ac.uk; 7Manchester Centre for Health Psychology, University of Manchester, Oxford Road, Manchester M13 9PT, UK; tracy.epton@manchester.ac.uk (T.E.); chris.armitage@manchester.ac.uk (C.J.A.); 8Centre for Behavioural Science and Applied Psychology, Sheffield Hallam University, Sheffield S10 2BQ, UK; m.arden@shu.ac.uk; 9School of Medical Sciences, University of Manchester, Stopford Building, Manchester M13 9PT, UK; jo.hart@manchester.ac.uk (J.H.); lucie.byrne-davis@manchester.ac.uk (L.M.T.B.-D.); 10School of Psychology, University of Sussex, Falmer BN1 9QN, UK; j.drury@sussex.ac.uk; 11North Yorkshire County Council, County Hall, Northallerton, North Yorkshire DL7 8DD, UK; eleanor.whittaker@northyorks.gov.uk; 12Public Health Wales, 2 Capital Quarter, Tyndall Street, Cardiff CF10 4BZ, UK; LLewis@somerset.gov.uk; 13Department of Behavioural Science and Health, Institute of Epidemiology and Health Care, University College London, London WC1E 6BT, UK; e.mcbride@ucl.ac.uk; 14Centre for Behaviour Change, University College London, 1-19 Torrington Place, London WC1E 7HB, UK; p.chadwick@ucl.ac.uk; 15Laboratory for Stress and Health Research, University of Leeds, Leeds LS2 9JT, UK; D.B.OConnor@leeds.ac.uk; 16Manchester University NHS Foundation Trust and NIHR Greater Manchester Patient Safety Translational Research Centre, Manchester M13 9PL, UK

**Keywords:** behavioural science, health psychology, consensus, COVID-19, rapid review, expert guidance, TRICE

## Abstract

Background: Public health emergencies require rapid responses from experts. Differing viewpoints are common in science, however, “mixed messaging” of varied perspectives can undermine credibility of experts; reduce trust in guidance; and act as a barrier to changing public health behaviours. Collation of a unified voice for effective knowledge creation and translation can be challenging. This work aimed to create a method for rapid psychologically-informed expert guidance during the COVID-19 response. Method: TRICE (Template for Rapid Iterative Consensus of Experts) brings structure, peer-review and consensus to the rapid generation of expert advice. It was developed and trialled with 15 core members of the British Psychological Society COVID-19 Behavioural Science and Disease Prevention Taskforce. Results: Using TRICE; we have produced 18 peer-reviewed COVID-19 guidance documents; based on rapid systematic reviews; co-created by experts in behavioural science and public health; taking 4–156 days to produce; with approximately 18 experts and a median of 7 drafts per output. We provide worked-examples and key considerations; including a shared ethos and theoretical/methodological framework; in this case; the Behaviour Change Wheel and COM-B. Conclusion: TRICE extends existing consensus methodologies and has supported public health collaboration; co-creation of guidance and translation of behavioural science to practice through explicit processes in generating expert advice for public health emergencies.

## 1. Introduction

Public health emergencies demand that experts provide rapid guidance to governments, scientists, practitioners, the media and the public. To maximise the effectiveness of this guidance, expert advice needs to be evidence-informed, consistent and credible. The speed at which public health emergencies develop, however, means that it can be difficult to ensure that multiple experts disseminate knowledge with a unified voice. Conflicting messages can be delivered inadvertently even when differences in opinion may be minimal. One way of overcoming perceived differences in opinion is to gain expert consensus, which should be based where possible on scientific evidence. The present paper describes a new Template for Rapid Iterative Consensus of Experts (TRICE) that can help improve collaborative public health expert responses to ongoing and future public health emergencies, ensuring the best use of scientific evidence from a unified voice and facilitating effective dissemination and translation. TRICE evolved from our behavioural science response to the COVID-19 pandemic. The core principles and features of our response presented here provide key considerations for the rapid synthesis of expert guidance.

### 1.1. Behavioural Science and the COVID-19 Pandemic

Behavioural science is an umbrella term that covers disciplines that deal with human actions, including the fundamental discipline of psychology, which is the science of human behaviour [1]. COVID-19 came from a virus with many ‘unknowns,’ however, the importance of behaviour and understanding influences on behaviour was clear from early on in the pandemic [2]. With a number of protective behaviours to consider in the prevention of viral transmission and a multitude of influences on these behaviours, consensus was needed to enable the development of guidance to support decision-makers and public health teams on optimal courses of action.

### 1.2. Consensus Methods

Consensus is defined as general agreement, which allows differences to co-exist, and is a method of knowledge production or decision making [3]. Consensus-based methods limit the biases that may occur when bringing together experts by having a formalised method and standardised approach to generate and synthesise knowledge that acknowledges the limitations. The three main types of consensus methodology are the nominal group process, Delphi methodologies, and consensus meetings [4,5]. The nominal group process requires assembling a team, the silent collection of ideas, and the later prioritisation of ideas to address a problem [6]. Whilst the key advantage of the nominal group process is that it minimises bias through anonymity, there can be challenges of assembling the panel in terms of space, time and the limitations of face-to-face meetings. Online meetings can mitigate these issues to some extent. Similarly, the ideas produced by the nominal group process are influenced by the panel/groups rather than a pre-defined methodological or theoretical approach. The nominal group process can suffer from a lack of flexibility with pre-defined parameters for consensus and limited interaction in which to broaden the scope from the original idea.

In the Delphi methodology, individuals numerically rank solutions to a pre-defined problem independently of each other [7,8]. When all have been ranked, the consensus of the group or subgroups is presented back to the panel along with an invitation to amend their position if they wish. This can take several rounds. Advantages include not having any one voice or group dominate the conversation, but with limited research defining what consensus looks like, researchers can decide what ranking determines consensus and introduce bias. High-quality work should define consensus in advance. Finally, consensus meetings are meetings of experts in a field to create a multi-disciplinary solution. It usually commences with a synthesis of the best available evidence, often a systematic review. Experts discuss the evidence in a face-to-face setting, determining the appropriate solution. Advantages include buy-in from experts who are likely to benefit from the outcomes and the use and scrutiny of evidence. However, evidence can take time to accumulate prior to the meeting and in the case of consensus development conference meetings, bias may occur due more vocal experts dominating the discussion [5].

### 1.3. Template for Rapid Iterative Consensus of Experts (TRICE)

Common to existing methods of gaining consensus is the slow pace at which consensus is generated. Each method requires time and planning. In public health emergencies, such as a global pandemic, there is a clear need for a rapid and pragmatic way to gain expert consensus that minimises bias and maximises the use of evidence and expertise. TRICE is a new 8-stage approach that brings structure, peer-review, and consensus to the rapid generation of expert advice. In the following sections, we describe how TRICE extends and accelerates existing consensus methodologies and report worked examples using our behavioural science response to the COVID-19 pandemic. The template will support public health collaboration and development through explicit processes in generating expert advice for public health emergencies. One key feature of TRICE that distinguishes it from existing consensus approaches is the explicit use of an agreed theoretical and epistemological approach that allows for rapid shared understanding and a unified voice.

## 2. The TRICE Method, Participants and Outputs

### 2.1. Stage 1: Identify Need

Table 1 provides an overview of the 8 stages using the TRICE method and examples from this work. The first stage of TRICE was to identify the specific need for information based on expert consensus. In public health emergencies, flow of information between groups with varying expertise such as academics, policymakers and practitioners to address issues in a timely manner may be hindered by structural communication barriers, that is, where connections do not exist between organisations and is often seen in healthcare settings with isolated clusters in need of connectivity [9].

In response to the COVID-19 pandemic, a small group from the British Psychological Society’s (BPS) Division of Health Psychology (JH, LMTB-D, AMC) met on 23 March 2020 to discuss a strategic response in recognition that behavioural science was key to managing the global pandemic. The need for a collective response was highlighted by requests from members of the Division of Health Psychology (academics, practitioners and those ‘in-training’), public health colleagues and the media. In parallel, following a series of strategic discussions and Twitter exchanges, the BPS convened a core COVID-19 coordinating group of 19 experts (including AMC, DBOC and JD) who first met on the 25 March 2020 to respond to the pandemic from a psychological perspective. Participants in SPI-B (Scientific Pandemic Influenza Group on Behaviours: the behavioural science subgroup of SAGE: the Scientific Advisory Group on Emergencies) sat within this core coordinating group and highlighted the need for a rapid, evidence-based response to government calls for support and evidence. The group was tasked with leading on different areas (https://www.bps.org.uk/coronavirus-resources/coordinating-group) (accessed on 13 August 2021) where psychology could support the COVID-19 response. One of these areas was Behavioural Science and Disease Prevention. A taskforce was convened (developed and led by AMC) to produce public health guidance to meet the needs of public health professionals and policymakers responding to the COVID-19 pandemic. The aim of this BPS COVID-19 Behavioural Science and Disease Prevention (BSDP) Taskforce was to provide behavioural science expertise and resources, at speed, to those requesting it, including SPI-B; as well as to wider public health agencies and local authorities to support the COVID-19 response.

### 2.2. Stage 2: Evaluate Capacity and Identify Stakeholders

Following the emergence of a public health threat, it is important that expert guidance is credible and trustworthy [10,11]. One way of achieving this is to involve credible institutions, such as learned societies and professional bodies early in the process. It might be that such credible institutions provide a rallying point with existing capacity and collation for experts to provide input. This can offer a shortcut to the identification of relevant experts and access to resources for wide-scale dissemination.

The BPS was the host of the BSDP Taskforce, all members were invited by AMC to represent key stakeholders and areas of expertise, including BPS Division of Health Psychology (DHP), BPS COVID-19 core coordinating group, the Behavioural Science and Public Health Network (BSPHN), the Health Psychology Exchange (HPX), University College London (UCL) Centre for Behaviour Change (CBC), the Behavioural Science Consortium, SPI-B (representing behavioural science and equality, diversity and inclusion), local authorities to include health and social care services, BPS Division of Clinical Psychology Public Health Prevention Taskforce, the Psychological Professions Network (PPN), BPS Board of Trustees, BPS Research Board and the BPS policy team. Representatives liaised with public health teams across the whole of the UK, ensuring that the work of the taskforce had relevance in England, Scotland, Northern Ireland and Wales, and took into account regional variations and ethnic minority communities. These individuals were invited as taskforce members to ensure that needs assessments could be identified swiftly, to assist co-creation of guidance with the target audiences, and to ensure wide-scale expertise, engagement, dissemination and translation. All member affiliations were supportive of their involvement and contribution, which was provided on a voluntary basis.

Capacity to respond at speed to the COVID-19 pandemic, at times needed a larger pool of expertise and resource. Drawing on previous successes from the development of The Change Exchange [12] and the BSPHN; [formally the Health Psychology in Public Health Network] [13,14] that have mobilised health psychology volunteers and created a platform to share best practice between behavioural science and public health, the Health Psychology Exchange (HPX) was developed as a platform for managing working groups and volunteers to support with requests for evidence-based rapid reviews, and to communicate with volunteers at speed. The founders and core members of the HPX (JH, AMC, LMTB-D) were also core members of the BPS DHP (who at the time were past, current and future DHP Chairs, respectively), acting as a conduit between the core stakeholder group (BPS BSDP Taskforce) and the voluntary collaborative of experts (HPX). To ensure inclusivity, HPX was not bound by the governance of one society or organisation and instead, all members of the HPX collectively agreed upon the use of open science principles and collaborative working.

### 2.3. Stage 3: Recruit to the Working Group

Knowledge held by members of a consensus group was seen as a source of novel information and innovation [9]. Each expert occupies positions within their professional network and can facilitate access to novel information, resources or transfer of knowledge across organisational and country boundaries. Structurally diverse consensus groups and collaboratives can increase the flow of information as experts are geographically sparse but intellectually connected. Where experts have a bridging role that spans across professional boundaries, e.g., public health practitioners with expertise in applied health psychology, including more than one expert, can minimise demands and bottlenecks in information flow from experts who risk being overloaded by others’ reliance on them [9].

Depending on the nature of the emergency, it is likely that guidance on multiple aspects will be needed, and it is useful for one named individual to lead a document with input from other experts. Most expert advice is given in the context of other competing demands (e.g., from principal employer, personal life), and thus, a shared load model in which not every expert needs to comment on every document provides flexibility and sustains momentum. Due to the volume of experts and differing leads for each document, it is valuable to foster a ‘natural’ peer-reviewing process, whereby those less involved in the core writing group become critical friends to draft documents. Documents published by the BPS had two sets of writing groups. Those who created the document based on meeting discussions of need, and those who were not part of the core writing groups who acted as peer-reviewers within the taskforce. Further, at least two members of the BPS policy and publication team also peer-reviewed each document published by the BPS. These latter reviewers were independent of the taskforce and not included in the authorship lists. Reviewer comments and editorial changes were made prior to final publication for all documents.

The COVID-19 Behavioural Science and Disease Prevention Taskforce was created with 13 initial members, expanding to 17 experts (*n* = 12 female; 5 male), with GWS, VS, TE and AK joining shortly after conception to ensure representation across the four-nations of the United Kingdom (UK), expertise in rapid reviews and equality, diversity and inclusion. Experts were situated in roles within academia (*n* = 11), public health (*n* = 2), the NHS (*n* = 1) and related to policy (*n* = 3). Collectively, expertise covered a diverse range of public health domains, including physical activity and sedentary behaviour, smoking cessation, dietary intake, alcohol and drug consumption, medication adherence, health service uptake, health professional behaviour, collective behaviours in emergencies and the role of culture on health. All held expertise in behavioural science and behaviour change. To ensure guidance could be useful across the UK, participants included those working in England, Scotland, Wales and Northern Ireland and across regional divides (e.g., north and south of England). Two were providing advice to Public Health England (one was directly employed), one was employed directly by Public Health Wales and two provided advice directly to Public Health Agencies in Northern Ireland and Scotland via the NHS and University sectors. Two members stood down from the taskforce towards the end of 2020, one was the policy lead for the BPS who left their role to move to another post elsewhere. The other was a clinical psychologist leading a BPS Clinical Psychology Public Health Prevention Taskforce who had stood down from their role and felt unable to contribute to the work of the Behavioural Science and Disease Prevention taskforce as they would have liked due to clinical commitments. Both members who stood down, felt it right to not continue to be included in the published outputs. Details of the core 15 members, can be found in Table 2 and was published at the onset for transparency (https://www.bps.org.uk/coronavirus-resources/coordinating-group/behavioural-science) (accessed on 13 August 2021).

The core group was assisted by three wider advisory groups of public health scientists, practitioners and those working in government: (1) the HPX (n = 155; led by JH, LMTB-D, AMC and structured using the Open Science Framework [OSF]); (2) the HPX Public Health Forum (n = 38; led by LJML and EW), proposed and supported by the BSDP Taskforce; (3) the BSDP Health Behaviour Working group (n = 15; led by AMC and GWS). Members from these groupings were also members of the BSDP main taskforce to facilitate rapid knowledge exchange between the groups. This type of brokerage role is known to facilitate transfer of knowledge and coordinate efforts across a collaborative [9]. The BSDP Taskforce lead (AMC) was a member of all groupings and aimed to attend all meetings to ensure consistency and dissemination of activities across each work-stream.

A total of eight (AMC, GWS, MAA, DBOC, CJA, JD, JH, TE) were active serving members of editorial teams for peer-reviewed publications and thus exposed to diverse methodologies, topics and areas of knowledge acquisition. All members agreed to adhere to standard good practice ground rules for meetings, including being on time, being open-minded and engaging in active listening. In a departure from standard practice, members agreed that all participants should be named authors on generated guidance that had been discussed in their presence during email exchanges and in meetings, regardless of the size of their contribution, unless they wished to recuse themselves or that they could not take responsibility for the content. Where rapid reviews were commissioned, there was a shared commitment to seeing them through to peer-reviewed publication, again to enhance the credibility of the output. On these occasions, a wider team was enlisted from the HPX to produce the review at speed while maintaining scientific integrity.

All members of the working groups agreed to use a single theoretical framework to standardise the approach and reduce conflicting messaging. The work was guided by the use of the Behaviour Change Wheel [15,16]. This framework has been advocated and used extensively within a public health setting [17] and is designed to assist with the development of behaviour change interventions. As the virus that causes COVID-19 can largely be mitigated by changes in behaviour (e.g., hand hygiene, wearing face coverings, physical-distancing) [18,19,20], this was deemed the most appropriate framework to use. Comparable with the ‘standard model’ in physics, it provided a shared theoretical framework and language within which the taskforce participants could operate that expedited and unified the process. The Behaviour Change Wheel contains several components useful to developing a shared vision of what to include in expert guidance. For example, it contains the capability, opportunity, motivation-behaviour (COM-B) model that condenses vast behaviour change literature, theories and frameworks, and facilitates recommendations for a whole-system approach, considering factors that influence behaviour on a micro, meso and macro level. Data from 2025 adults, gathered from a survey launched 52 days after the first confirmed case of COVID-19 in the United Kingdom, provided good evidence for the ability of COM-B to explain disease prevention behaviours [21]. These included: touching eyes or mouth, cleaning hands (with soap and water or alcohol-gel), disinfecting home surfaces, and covering nose and mouth with a tissue or sleeve when coughing or sneezing.

The first piece of guidance produced by the BSDP taskforce on 14 April 2020 displayed a COM-B ‘roadmap’ [10] as a template of core open-ended questions for consideration when performing a COM-B behavioural diagnosis for disease prevention behaviours related to COVID-19. This template, based on AMC’s Intervention Design, Delivery, Evaluation and Adoption System (IDDEAS) training model for behaviour change [22], was then used for subsequent guidance documents tailored to specific health behaviours.

### 2.4. Stage 4: Iteration Task 1

In a departure from traditional expert consensus procedures, TRICE advocates that:(1)Priorities for guidance are informed by the needs of stakeholders;(2)The expert group themselves generate, distil and judge the evidence-based content to use in guidance documents(3)All work uses a shared theoretical framework(4)Guidance is checked with stakeholders prior to publication

The taskforce specified the core ‘problem’ to be addressed, specifying the relevant behaviour to target for each guidance document, as well as the target audience for each piece of work. The maximum length of each document was specified, based on feedback from key stakeholders (e.g., government officials, those working in policy teams); short and concise was recommended. For the development of each guidance document, the expert group drew on a range of materials, including existing evidence on related coronavirus pandemics such as SARS and MERS, authors pre-COVID-19 expertise in their respective areas, commissioned rapid reviews, emerging empirical evidence, theoretical mapping to novel coronavirus challenges and data from UK government and international sources (e.g., SAGE, Independent SAGE, Office for National Statistics, World Health Organisation).

Technology enabled the implementation and sharing of tasks. Documents were shared via email, moving to Googledocs after the first guidance document, providing editing privileges to all authors to combine materials and comments at speed to facilitate simultaneous real-time collaboration. The wider HPX group used an Open Science Framework (osf.io) as a platform to share documents, including those produced by the BSDP taskforce, that might be useful to others facing similar challenges at a local, national, or international level, and HPX email list (managed by LMTB-D) for rapid requests for information. The Taskforce later moved to the use of a dedicated area of Microsoft Teams to host all materials and continued to work on shared documents to ensure document version control.

### 2.5. Stage 5: Iteration Task 2

One advantage of TRICE is that it allows experts to consider what are the commitments involved in contributing to expert guidance. Depending on the nature of the emergency, it is likely that numerous meetings will be required in a short period of time, and to ensure attendance, it is useful to inform experts of the nature of the iterative task.

In the present case, experts were initially invited to attend weekly hour-long face-to-face virtual meetings (via Zoom) to highlight areas of need, discuss requests from public health colleagues (e.g., via the HPX Public Health Forum) and pre-emptive areas of concern to generate topics for guidance. Meetings were scheduled for the same time and day as agreed by the taskforce, which was weekly for a large part of the first year, moving to fortnightly and then monthly. A member of the core group volunteered to lead each piece of guidance, which they first drafted and circulated to the core group for comment. Use of the COM-B approach and road map allowed for a shared ‘template’ for guidance development, making draft preparation and iteration more efficient and standardised the dissemination format without compromising message delivery formats. Consensus was achieved through document comments and discussion at meetings. Experts were encouraged to discuss with the wider advisory group and their professional networks in-between meetings to incorporate the latest need, evidence and feedback. All additional contributors were acknowledged.

### 2.6. Stage 6: Final Draft Review

As noted above, not every expert needs to comment on every document using the TRICE method, however, to retain a unified voice, it is valuable if the taskforce lead is involved in all core writing groups to maintain consistency. AMC performed this function in the present example. Documents were further circulated for comment and discussed via the wider BPS core COVID-19 Coordinating Group once in final draft. All documents were reviewed by at least two members of BPS staff from the policy and publication team prior to finalisation. It is valuable to have multiple stakeholders and, ideally, non-experts review final documents before publication as part of the peer-review process.

### 2.7. Stage 7: Implementation

Implementation of expert opinion requires not just the production of documents, but also dissemination through multiple channels of communication, including via face-to-face meetings with stakeholders, press releases, media appearances, and crucially feeding into the key policy making organisations (e.g., SPI-B and public health agencies in the case of the COVID-19 pandemic), to Directors of Public Health and practitioners in public health.

As of spring 2021, the core taskforce had produced 18 peer-reviewed guidance documents, presented in Table 3 alongside four rapid systematic reviews, an interim evaluation and key information in relation to the process of development. Days until publication differed markedly, from 4 days to 156 (including weekend days). Those that took longer were often due to issues of production through the host organisation (BPS), who were producing guidance for many other work-streams as well as the BSDP Taskforce. What is key is that the median number of drafts was 7 per guidance document, achieved with a median of 18 experts involved. This is important because expert groups are almost always convened on a voluntary basis. Providing people with an idea of how much resource is needed and for how long it will be needed are important drivers of prosocial behaviour [23].

### 2.8. Stage 8: Evaluation

Evaluation is a crucial, ongoing, phase in TRICE. It is important to assess the use and impact of guidance produced for policymakers, public health practitioners and the public. This can be achieved through primary research designed to assess awareness and uptake of materials, sustained use of materials and shifts in policy, as well as citations in policy, media and academic circles. Initial evaluations have begun to sense check the work of the BSDP Taskforce, how it has been used and the impact it has had. This includes case studies [1] such as that of North Yorkshire County Council whose representative stated: *“The guidance* [10] *is included in both the public health and corporate COVID-19 communications plan, meaning that all members of staff who have responsibility for communications across our council have had access to the guidance, and we have discussed how we can use it to make our communications more likely to change behaviour. Our communications staff have found the guidance invaluable and welcomed the way that it was accessible, easy to follow, and ‘not too academic.’”* Another example of how this guidance has been used to optimize public health campaigns comes from the City of Wolverhampton Council who said: *“We focus on a community approach throughout our ‘Stay Safe Be Kind’ campaign which reinforces the idea that individuals should look after each other by minimising the ‘I’ and focusing on the ‘we’”*. These examples draw on specific recommendations from the first guidance document produced by the Taskforce [10]. Finally, the Aneurin Bevan Local Public Health Team have highlighted that: *“We are using the nine recommendations in the guidance* [10] *as a way of quality assuring and optimising our public facing communications ahead of contact tracing commencing in Gwent”*.

The work of the group [10,24] has further been used as evidence for SPI-B (https://assets.publishing.service.gov.uk/government/uploads/system/uploads/attachment_data/file/888750/7b._20.04.27_SPI-B_behavioural_science_notes_on_symptom_vs_test_based_approaches_S0260.pdf) (accessed on 13 August 2021), and translated into Japanese (https://psych.or.jp/special/covid19/Behavioural_science_and_disease_prevention/) (accessed on 13 August 2021) [10]. The self-isolation guidance [30] has been used verbatim by Directors of Public Health in public-facing videos promoting top tips to enable self-isolation (https://www.youtube.com/embed/QILZienYmls) (accessed on 13 August 2021, and in recommendations by Public Health Wales (https://phw.nhs.wales/publications/publications1/self-isolation-confidence-adherence-and-challenges-behavioural-insights-from-contacts-of-cases-of-covid-19-starting-and-completing-self-isolation-in-wales/) (accessed on 13 August 2021). The NHS have created a version of the guidance on what to do after the first dose of the vaccine [39] (http://www.airedale-trust.nhs.uk/wp/wp-content/uploads/2021/05/NHS-after-your-first-vaccine-guidance.pdf) (accessed on 13 August 2021), and with the support of taskforce members, Lifebuoy have used the hand hygiene guidance to assist in the development of a wide-scale school hand hygiene programme (http://www.lordblytonprimaryschool.co.uk/wp-content/uploads/2020/10/07_-Leaflet_for_Parents_Guardians.pdf) (accessed on 13 August 2021). Future evaluations will draw on formal research methods and reporting to fully evaluate the use of the series of guidance documents and the impact they have had on the behaviours of policymakers, public health officials and the general population.

## 3. Discussion

This paper highlights important considerations for future investment and a new expert consensus approach that is better suited to rapid responsiveness than traditional consensus approaches. TRICE draws on established consensus methods such as the Delphi, nominal and consensus meeting methodologies by defining the theme or problem to be examined and drawing on experts to achieve consensus through an iterative process. The process of knowledge production in TRICE differs from other methods in the preparation for consensus, the lack of anonymity and the suitability for online collaboration. TRICE advocates a collaborative, open and consultative 8-stage approach, which draws on various members holding bridging roles between stakeholders (e.g., academic and public health) and brokering/boundary-spanning roles for rapid mobilisation and critique of shared knowledge and expertise to address the problem. Using this approach, teams of approximately 15–18 experts can produce high-quality, evidence-based guidance at speed.

This can be facilitated by regular (starting at weekly) face-to-face meetings, having one lead per guidance document, and a lead for the oversight group. Target audiences, length of document and strategic goals in terms of impact should be agreed early on and stakeholder involvement is integral to the process from start to finish. Unlike consensus meetings, which may require considerable preparation and evidence assembly prior to meetings [3], the TRICE approach utilises the multi-stakeholder experience to produce the evidence at speed cognisant of the practicalities of application to the problem. Documents had approximately seven iterations to reach publication quality in which the evidence underpinning recommendations can be challenged, clarified and resolved. To minimise bias, there should be an effective system in place for internal expert review and external review by at least two non-experts.

Taking this rapid consensus approach enabled a collective psychologically informed response to the COVID-19 pandemic, at speed, during a time when expertise was called upon from across the UK nations. This work, and the continued request for support, has highlighted the important contribution of behavioural science, and more specifically health psychology and behaviour change, in the pandemic response [19,45,46]. This was a novel, efficient and effective approach, building on existing collaborations, and new partnerships with a group who shared the enthusiasm and drive to support the global response.

### 3.1. Strengths

TRICE provides a framework to guide rapid mobilisation of experts, which draws on existing and new partnerships. The validity of group consensus is higher than individual knowledge development [47] and the TRICE framework outlines a clear process of obtaining consensus at each stage, which can be replicated in other settings. The real-time collaboration on documents from people with a range of expertise and the ability to produce high-quality guidance very rapidly at very low financial cost was a positive achievement. It is important to acknowledge the informal and voluntary nature of this work. Although not formally resourced, it represented a considerable contribution to and investment in the health and wellbeing of society by a range of public health agencies, universities, practitioners and a national learned society and charity, the British Psychological Society. Different perspectives from experts working in different areas but who apply the same theoretical frameworks to health issues, facilitated the efficient application of theory to the novel challenges posed by COVID-19. This highlights the importance of having an agreed, widely used and understood theoretical approach. Furthermore, two of the authors (PC, AMC) were members of the UCL Centre for Behaviour Change, where the chosen theoretical approach, the BCW had been developed, which facilitated its appropriate use. The field of behavioural science also benefits generally from existing excellent collaborative communication and working links, and the ability to draw on international high-quality evidence, which was very relevant to provide guidance during the COVID-19 pandemic. While the target audience for this work was predominately decision-makers and public health teams, there is scope to use this approach to generate further guidance specifically for the general population.

The UK has a system of devolved delivery of health services. This means that healthcare policies and implementation can vary, thus guidance requires adaptation to context. Unified perspectives from across the UK ensured the guidance was adaptable to the four nations and reflected commonalities in priorities and behaviours to target. This was facilitated by identifying gatekeepers in each of the different nations who understand and are connected to local public health systems and understand the different contexts. Recommendations, although UK-specific, can, and have been utilised by other countries.

### 3.2. Challenges

The challenges experienced mirror challenges of other voluntary schemes during the pandemic where volunteer projects have had an impact but at the cost of longer-term fatigue and over-work among volunteers if they are not supported with resources [48]. Voluntary contributions on top of already high workloads meant that time was limited and often spilled outside of ‘working hours,’ making the process challenging. This was commonly seen across psychology during the pandemic [49]. This is not a sustainable model and funding for rapid consensus approaches should be considered and made available in the future; a call to action made both by this group [41] and others [50]. Financial resourcing is also likely to speed up processes further, as those involved voluntarily could dedicate more time if the role was part of paid work. Relying on the goodwill of those involved risks burnout and should be avoided. This requires establishing clear limits and boundaries on the part of those involved in the voluntary ‘acute’ phase on responding to a global health crisis. However, this paper highlights the likely time commitments involved, from the likely number of experts, while contributing in a voluntary capacity. An investment in behavioural science, and specifically health psychology, in all public health teams as a norm [41] would further expedite future consensus work.

Management of documents and iterations that were in some instances shared via email necessitated a good tracking system, and at times multiple versions needed to be collated. Using Googledocs overcame this issue, and the Taskforce are now using Microsoft Teams to host all materials.

Delays also occurred in the early phase of the linked series of health behaviour documents, while agreeing on a template for the content and presentation of material that would be most helpful to the target audience, who were in this case, public health officials and policymakers. It was also reliant on six documents (public health roadmaps for physical activity, sedentary behaviour, eating behaviour, alcohol consumption, stopping smoking and sleep hygiene) being ready to be launched together, each with different lead authors, rather than working on standalone documents. They also included a wider writing group. This led to a longer production time to ensure there was a coherent consensus running through all six documents. We wanted to streamline the content to have a similar format as much as possible to make is easier for our target audience (e.g., public health officials) to use the content and understand the process of behaviour change (e.g., identify problem and target population, specify target behaviour to deal with problem, highlight COM-B drivers of behaviour, highlight policy options to influence change), and provide links for support. Sending documents to production also led to some delay, with several editorial corrections needed to ensure a seamless series and approximately a two-week turn-around from draft document to published material. This was due to the sheer pace and volume at which the BPS production team were working, and this would be another area where greater resource would be beneficial in future pandemics.

## 4. Conclusions

The TRICE approach used through the BPS COVID-19 Behavioural Science and Disease Prevention taskforce, and the guidance documents produced, have been disseminated widely with global impact. This approach to the development of such material and the way in which it has been presented extends current methodologies in this area and can be used as a template for future work. The taskforce will now focus their attention on an evaluation of the impact of these documents on the global efforts to reduce the number of COVID-19 cases.

## Figures and Tables

**Table 1 ijerph-18-10255-t001:** Summary of the Template for Rapid Iterative Consensus of Experts (TRICE) and examples of the use of TRICE during the COVID-19 pandemic.

Target	Process	Considerations	Example
1. IDENTIFY NEED	Scope problem, undertake a needs assessment Identify target population(s) Align with policy drivers Identify system gaps	Policy driven and/or population driven (top-down, bottom-up) Commissioned and/or requested from a range of stakeholder groups	Numerous requests from stakeholders (scientists, public health practitioners, policymakers, the public) for a unified voice in the initial stages of the global COVID-19 pandemic
2. EVALUATE CAPACITY AND IDENTIFY STAKEHOLDERS	Identify support and obtain buy-in from: ¯ Academics/scientists ¯ Public health practitioners ¯ Service providers/NHS ¯ Policy leaders	Include credible, trustworthy organisations Functional role agreed Identify efficient and inclusive communication processes Negotiate commitment (moral, resource)	Representatives from the BPS Division of Health Psychology created a national network (Health Psychology Exchange, HPX) to facilitate knowledge exchange and perform rapid scoping reviews Creation of the core COVID-19 coordinating group within trustworthy organisation (British Psychological Society) and the COVID-19 Behavioural Science and Disease Prevention Taskforce. Identify representative stakeholders with relevant expertise for shared agenda to join the taskforce
3. RECRUIT TO WORKING GROUP	**Criteria (Essential)** Invite relevant experts Ensure geo-political and socio-demographic diversity Assess time commitment available Agree on core principles and frameworks	**Criteria (Desirable)** Agree leadership Identify shared theoretical framework Enable open communication Link to supportive networks and stakeholder organisations	Inclusion of representatives from the following: BPS Division of Health Psychology; Behavioural Science and Public Health Network (BSPHN); Health Psychology Exchange (HPX), SPI-B; NHS; Psychological Professions Network (PPN), Public Health, Academia, UCL Centre for Behaviour Change, Behavioural Science Consortium Wider BPS: social psychology, clinical psychology, sport and exercise psychology, community psychology Agreed on four Nations (England, Scotland, Wales, Northern Ireland) approach Agreed to use Behaviour Change Wheel (BCW) and COM-B (capability, opportunity, motivation–behaviour) model Agreed to actively consider issues related to equality, diversity and inclusion
4. ITERATION: TASK 1	Identify ‘problem’ for expert guidance to target Identify and specify core behaviour(s) linked to problem to target Identify target audience for expert guidance Identify task lead Define parameters/criteria Collate evidence Apply theoretical framework Synthesise evidence Draft the report	Shared leadership Experience of applying relevant theoretical framework Shared contribution to evidence collation and synthesis	Created a working overview document of tasks with identified ‘problem,’ target behaviour to mitigate problem, target audience for guidance, named lead(s), core writing group and ‘critical friends’ for peer-review, links to working draft guidance documents and progress Shared tasks; writing, reviewing, critique according to expertise and capacity Worked to clear timeframes Had one person with overall oversight of all activities (AMC) Detailed minutes kept and shared with group with overview document updated at each meeting All documents kept in a central location for all to access (Google Drive and Microsoft Teams)
5. ITERATION: TASK 2	Stakeholder review and feedback for: relevance, usability, contextual applicability, cost effectiveness, equality, diversity and inclusion Revise and re-draft content as appropriate	Peer review via whole group and expert networks Rapid turn-around End-user review Sense-check, usability check, language check	Discussed guidance documents with stakeholder groups—e.g., World Health Organisation (WHO) consultants, practitioners and public health consultants via the HPX Public Health Forum, local authority colleagues and directors of public health
6. FINAL DRAFT REVIEW	Rapid review of the final report via commissioners, networks, stakeholder groups Sign off by stakeholders	Reviewed by independent non-experts Shared responsibility/authorship in taskforce group	Sign-off from writing group, core taskforce group Approval from BPS Policy team Approval from BPS comms team Approval from BPS core COVID-19 coordinating group members
7. IMPLEMENTATION	Website hosting Stakeholder dissemination Media dissemination Open Science dissemination Webinars and learning events	Identify core spokespeople (including document lead for media engagement) Generate press release with key messages Have a consistent message from the group lead to run through all outputs Translate to other languages for wider dissemination	All guidance documents hosted on the BPS website or open-access peer-reviewed websites Lead author and taskforce lead (AMC) identified as media spokespeople for all guidance documents BPS wrote a press release for each guidance document, with support from stakeholder press offices (e.g., universities) Core message for each release linked to initial guidance (e.g., *‘Use behavioural science to Combat COVID-19 together’* Disseminated via: Public Health England Lunch and Learn webinar; Behavioural Science and Public Health Network hub events organised by Health Education England and Public Health England; BPS Division of Health Psychology COVID-19 webinar; Psychological Professions Network webinar; British Psychological Society annual conference webinar; events for commissioners Sent to global colleagues where some were translated to different languages for wider access internationally
8. EVALUATION	Monitor media exposure, use of guidance, citations in policy documents Research mechanisms of impact (e.g., qualitative interviews, surveys) Feedback on TRICE process from expert members	Acquire research funding for evaluation Gain feedback from stakeholder groups of use	Collation of impact and media exposure Focus groups with key stakeholders (e.g., via the HPX Public Health Forum) on use and impact Informal and formal peer review of outputs

**Table 2 ijerph-18-10255-t002:** Affiliations and biographies of each core member of the British Psychological Society COVID-19 Behavioural Science and Disease Prevention (BSDP) Taskforce.

BSDP Taskforce Member	Affiliation and Brief Biography
Professor Angel Marie Chater	Registered Health Psychologist (HCPC) and Professor in Health Psychology and Behaviour Change, University of Bedfordshire, UK. Director of the Institute for Sport and Physical Activity Research and Lead of the Centre for Health, Wellbeing and Behaviour Change, University of Bedfordshire. Associate to the University College London Centre for Behaviour Change. Chair of the British Psychological Society’s (BPS) Division of Health Psychology (Jul 2019–Jun 2021). Co-founder of the Behavioural Science and Public Health Network (BSPHN) and inaugural Chair. Co-founder of the Health Psychology Exchange (HPX). Consulting Editor of Health Psychology and Behavioural Medicine and Behavioural Science and Public Health journals. Lead of the BPS COVID-19 Behavioural Science and Disease Prevention Taskforce.
Dr Gillian W Shorter	Lecturer in Clinical Psychology, Queen’s University Belfast. Co-Director of Drug and Alcohol Research Network, Queen’s University Belfast. Public Health Agency Northern Ireland Behaviour Change Cell, North South Alcohol Policy Advisory Board. New Strategic Direction Alcohol and Drugs Advisory Board member. Chair of Digital Interventions Interest Group (Health Informatics) in the MRC/NIHR Trials Methodology Research Partnership. Research Lead for the BPS Division of Health Psychology (2020–2021). Academic Editor/Editorial Board PLoS ONE, Drugs: Education, Prevention, and Policy. Invited member of the BPS COVID-19 Behavioural Science and Disease Prevention Taskforce.
Dr Vivien Swanson	Registered Health Psychologist (HCPC). Reader in Health Psychology and Professional Doctorate Programme Director at the University of Stirling, Scotland, UK. Programme Lead for Health Psychology Professional Practice, NHS Education for Scotland. Member and Past-Chair BPS Division of Health Psychology, Scotland, and member of BPS Qualifications Board (Health Psychology). Founding partner of the Change Exchange Hub at the University of Stirling. Invited member of the BPS COVID-19 Behavioural Science and Disease Prevention Taskforce.
Dr Atiya Kamal	Registered Health Psychologist (HCPC) and Senior Lecturer at Birmingham City University, UK. Participant of SPI-B, the behavioural science subgroup of SAGE, the Ethnicity sub-group of SAGE and the International Best Practice Advisory Group, which provides expert input to the analysis of international responses to the COVID-19 pandemic. Conference Lead for the BPS Division of Health Psychology. Founding partner of The Change Exchange hub at Birmingham City University. Invited member of the BPS COVID-19 Behavioural Science and Disease Prevention Taskforce.
Dr Tracy Epton	Lecturer in Health Psychology, University of Manchester, UK. Member of the Health Psychology Exchange responsible for coordinating crowd-sourced rapid reviews. Communications Lead for the BPS Division of Health Psychology (2020–2024). Invited member of the BPS COVID-19 Behavioural Science and Disease Prevention Taskforce.
Professor Madelynne A Arden	Registered Health Psychologist (HCPC) and Professor of Health Psychology at Sheffield Hallam University, UK. Director of the Centre for Behavioural Science and Applied Psychology and the Behavioural Science Consortium. Co-Editor of the British Journal of Health Psychology. Fellow of the Academy of Social Sciences. Co-chair of the Behavioural Science Hub of the Yorkshire and Humber Public Health Network. Invited member of the BPS COVID-19 Behavioural Science and Disease Prevention Taskforce.
Professor Jo Hart	Registered Health Psychologist (HCPC), Professor of Health Professional Education and Head of the Division of Medical Education, University of Manchester, UK. Co-founder of The Change Exchange and the Health Psychology Exchange. Works with Health Education England and Public Health England. Past Chair of the BPS Division of Health Psychology (Chair: 2017–2019). Fellow of the European Health Psychology Society and the Academy of Social Sciences. Principal Fellow of Advance HE. Invited member of the BPS COVID-19 Behavioural Science and Disease Prevention Taskforce.
Professor Lucie MT Byrne-Davis	Registered Health Psychologist (HCPC) and Professor of Health Psychology, University of Manchester, UK. Chair of the British Psychological Society’s (BPS) Division of Health Psychology (Jun 2021–Jun 2023). Chair of the European Health Psychology Society’s United Nations Committee. Psychological Professions Network North-West Workforce Council Board Member for health psychology. Co-founder of The Change Exchange and The Health Psychology Exchange. Fellow of the European Health Psychology Society and Principal Fellow of the Higher Education Academy. Invited member of the BPS COVID-19 Behavioural Science and Disease Prevention Taskforce.
Professor John Drury	Social Psychologist, University of Sussex, UK. Specialises in collective behaviour, including behaviour in emergencies and disasters. Advisor on UK government expert groups (since 2010) on public behaviour in emergencies, including participating in SPI-B and the behavioural subgroup of SAGE during the COVID-19 pandemic. Invited member of the BPS COVID-19 Behavioural Science and Disease Prevention Taskforce.
Dr Ellie Whittaker	Health Improvement Officer in the Public Health Team, North Yorkshire County Council. Co-chair of Health Psychology Exchange Public Health Forum. Health Psychology Champion on BPS Division of Health Psychology Committee (2021–2022). Invited member of the BPS COVID-19 Behavioural Science and Disease Prevention Taskforce.
Mrs Lesley J M Lewis	Health Psychologist in Training, Staffordshire University and Registered Public Health Practitioner (UKPHR). Behaviour change specialist, Public Health Wales. Co-chair of Health Psychology Exchange Public Health Forum. Assistant Publication Editor on Behavioural Science and Public Health Network Committee. Invited member of the BPS COVID-19 Behavioural Science and Disease Prevention Taskforce.
Dr Emily McBride	Registered Health Psychologist (HCPC) and Senior Research Fellow, Department of Behavioural Science and Health at University College London (UCL), UK. National Institute for Health Research (NIHR) Fellow. Policy Lead for the BPS Division of Health Psychology (Jul 2019–Jun 2022). Invited member of the BPS COVID-19 Behavioural Science and Disease Prevention Taskforce.
Dr Paul Chadwick	Registered Clinical and Health Psychologist (HCPC). Associate Professor of Behaviour Change, University College London Centre for Behaviour Change, UK. Lead for the Behavioural and Social Science Strategy in Public Health England. Invited member of the BPS COVID-19 Behavioural Science and Disease Prevention Taskforce.
Professor Daryl B O’Connor	Registered Health Psychologist (HCPC) and Professor of Psychology at the University of Leeds, UK. Deputy Chair of the BPS COVID-19 Core Coordinating Group and Co-lead of the BPS COVID-19 Research Priorities Group (2020–21). Chair of the BPS Research Board (2015–2021), BPS Trustee (2015–2021) and Chair of the European Federation of Psychology Associations (EFPA) Board of Scientific Affairs (2017–2021). Past Chair of the BPS Division of Health Psychology and BPS Psychobiology Section. Editor-in-Chief, Cogent Psychology and past joint Editor-in-Chief, Psychology and Health (2011–2019). Invited member of the BPS COVID-19 Behavioural Science and Disease Prevention Taskforce.
Professor Christopher J Armitage	Registered Health Psychologist (HCPC), Professor of Health Psychology and Research Director of the Manchester Centre for Health Psychology, University of Manchester, UK. Behavioural Science Research Lead for the NIHR Greater Manchester Patient Safety Translational Research Centre. Past Chair of the British Psychological Society Division of Health Psychology. Fellow of the Academy of Social Sciences. Associate Editor, Psychology and Health (2008-present). Co-lead of the BPS COVID-19 Research Priorities Group (2020–21). Invited member of the BPS COVID-19 Behavioural Science and Disease Prevention Taskforce

**Table 3 ijerph-18-10255-t003:** Outputs generated from the British Psychological Society COVID-19 Behavioural Science and Disease Prevention Taskforce and the process of rapid consensus for production.

Type of Document	Authors, Title and Link	Date Started	Date Published	Days to Publish (Incl. Weekend)	N of Experts	N of Meetings	N of Drafts Incl. Production
**Guidance** [10]	**Chater et al., (2020).***Behavioural science and disease prevention: Psychological guidance*. https://www.bps.org.uk/sites/www.bps.org.uk/files/Policy/Policy%20-%20Files/Behavioural%20science%20and%20disease%20prevention%20-%20Psychological%20guidance%20for%20optimising%20policies%20and%20communication.pdf (accessed on 13 August 2021)	29-03-20	14-04-20	16	13	2	7
**Evidence synthesis** [24]	**Thorneloe et al., (2020).***Scoping review of mobile phone app uptake and engagement to inform digital contact tracing tools for COVID-19.*https://psyarxiv.com/qe9b6/ (accessed on 13 August 2021)	21-04-20	25-04-20 (preprint)	4	17	6	2
**Guidance** [25]	**Arden et al., (2020).***Behavioural science and success of the proposed UK digital contact tracing application for Covid-19. * https://www.bps.org.uk/sites/www.bps.org.uk/files/Policy/Policy%20-%20Files/Behavioural%20science%20-%20digital%20contact%20tracing%20for%20Covid-19.pdf (accessed on 13 August 2021)	15-05-20	04-06-20	20	15	2	5
**Evidence-synthesis [linked]** [26]	**Ghio et al., (2020).***What influences people’s responses to public health messages for managing risks and preventing disease during public health crises? A rapid review of the evidence and recommendations.* https://psyarxiv.com/nz7tr/ (accessed on 13 August 2021)	04-06-20	13-07-20 (preprint)	39	23	15	4
[27]	**Keyworth et al., (2020).***What are the most effective public health messages for managing risks and preventing disease during public health crises?* Linked PROSPERO record. https://www.crd.york.ac.uk/prospero/display_record.php?RecordID=188704 (accessed on 13 August 2021)						
[28]	**Lawes-Wickwar et al. (2021).** A rapid systematic review of public responses to health messages encouraging vaccination against infectious diseases in a pandemic or epidemic. *Vaccines, 9*(2), 72. https://www.mdpi.com/2076-393X/9/2/72/htm (accessed on 13 August 2021)						
**Guidance/Thought piece** [29]	**Armitage et al., (2020).***Why simply asking people to self-isolate won’t cut it.* BPS Blog. https://www.bps.org.uk/blogs/bps-communications-team/why-simply-asking-people-self-isolate-wont-cut-it (accessed on 13 August 2021)	04-06-20	30-06-20	26	11	1	3
**Guidance** [30]	**Arden et al., (2020).***Encouraging self-isolation to prevent the spread of COVID-19.* https://www.bps.org.uk/sites/www.bps.org.uk/files/Policy/Policy%20-%20Files/Encouraging%20self-isolation%20to%20prevent%20the%20spread%20of%20Covid-19.pdf (accessed on 13 August 2021)	04-06-20	07-09-20	95	16	9	11
**Guidance** [31]	**Byrne-Davis, et al., (2020a).***Encouraging hand hygiene in the community.*https://www.bps.org.uk/sites/www.bps.org.uk/files/Policy/Policy%20-%20Files/Encouraging%20hand%20hygiene%20in%20the%20community.pdf (accessed on 13 August 2021)	26-06-20	24-07-20	28	18	4	11
**Guidance** [32]	**Byrne-Davis, et al., (2020b).***The Psychology of Hand Washing.* https://www.bps.org.uk/coronavirus-resources/public/handwashing (accessed on 13 August 2021)	01-07-20	06-08-20	36	18	6	6
**Guidance** [33]	**Chater, Abdin, Dryden, et al., (2020).***COVID-19 Public Health Road Map: Physical Activity.*https://www.bps.org.uk/sites/www.bps.org.uk/files/Policy/Policy%20-%20Files/Covid-19%20Public%20Health%20Road%20Map%20%E2%80%93%20Physical%20activity.pdf (accessed on 13 August 2021)	28-05-20	27-10-20	155 (linked series)	19	2	10
**Guidance** [34]	**Chater, Abdin, Shorter, et al., (2020).***COVID-19 Public Health Road Map: Sedentary Behaviour.*https://www.bps.org.uk/sites/www.bps.org.uk/files/Policy/Policy%20-%20Files/Covid-19%20Public%20Health%20Road%20Map%20-%20Sedentary%20behaviour.pdf (accessed on 13 August 2021)	28-05-20	27-10-20	155 (linked series)	18	2	10
**Guidance** [35]	**Whittaker, et al., (2020).***COVID-19 Public Health Road Map: Eating Behaviour.* https://www.bps.org.uk/sites/www.bps.org.uk/files/Policy/Policy%20-%20Files/Covid-19%20Public%20Health%20Road%20Map%20-%20Eating%20behaviour.pdf (accessed on 13 August 2021)	28-05-20	27-10-20	155 (linked series)	18	2	10
**Guidance** [36]	**Knowles, et al., (2020).***COVID-19 Public Health Road Map: Stopping Smoking.* https://www.bps.org.uk/sites/www.bps.org.uk/files/Policy/Policy%20-%20Files/Covid-19%20Public%20Health%20Road%20Map%20-%20Stopping%20smoking.pdf (accessed on 13 August 2021)	28-05-20	27-10-20	155 (linked series)	18	2	10
**Guidance** [37]	**Shorter, et al., (2020).***COVID-19 Public Health Road Map: Alcohol Consumption.* https://www.bps.org.uk/sites/www.bps.org.uk/files/Policy/Policy%20-%20Files/Covid-19%20Public%20Health%20Road%20Map%20-%20Alcohol%20consumption.pdf (accessed on 13 August 2021)	28-05-20	27-10-20	155 (linked series)	19	3	10
**Guidance** [38]	**Jenkinson, et al., (2020).***COVID-19 Public Health Road Map: Sleep Hygiene.* https://www.bps.org.uk/sites/www.bps.org.uk/files/Policy/Policy%20-%20Files/Covid-19%20Public%20Health%20Road%20Map%20-%20Sleep%20hygiene.pdf (accessed on 13 August 2021)	28-05-20	27-10-20	155 (linked series)	18	3	10
**Interim evaluation** [2]	**Chater et al., (2020).** Health psychology, behavioural science, and Covid-19 disease prevention. *Health Psychology Update, 29 SI*, 3-9	23-06-20	19-07-20	26	15	3	3
**Guidance** [11]	**Epton et al., (2020).***Delivering effective public health campaigns during COVID-19.* https://www.bps.org.uk/sites/www.bps.org.uk/files/Policy/Policy%20-%20Files/Delivering%20effective%20public%20health%20campaigns%20during%20Covid-19.pdf (accessed on 13 August 2021)	21-09-20	17-11-20	57	19	3	10
**Guidance** [39]	**Arden et al., (2021).***Guidance following first vaccination dose*. https://www.bps.org.uk/sites/www.bps.org.uk/files/Policy/Policy%20-%20Files/Guidance%20following%20your%20first%20vaccination%20dose.pdf (accessed on 13 August 2021)	10-02-21	01-03-21	19	15	2	7
**Guidance** [40]	**Epton et al., (2021).***Optimising vaccination uptake for COVID-19* https://www.bps.org.uk/sites/www.bps.org.uk/files/Policy/Policy%20-%20Files/Optimising%20vaccine%20uptake.pdf (accessed on 13 August 2021)	04-02-21	01-04-21	56	17	2	7
**Guidance/Briefing** [41]	**McBride et al., (2021).***Behavioural science investment needed to mitigate long-term health impacts of COVID-19* https://www.bps.org.uk/sites/www.bps.org.uk/files/Policy/Policy%20-%20Files/Behavioural%20Science%20Investment%20Needed%20to%20Mitigate%20the%20Long-Term%20Health%20Impacts%20of%20Covid-19.pdf (accessed on 13 August 2021)	11-01-21	16-06-21	156	15	5	4
**Evidence synthesis** [42]	**Epton et al. (2021).** Systematic review of interventions to promote the performance of physical distancing behaviours during pandemics/epidemics of infectious diseases spread via aerosols or droplets. https://psyarxiv.com/rn4vb/ (accessed on 13 August 2021)	15-01-21	13-06-21	149	30	5	3
**Guidance** [43]	**Hart et al., (2021).***Optimising physical distancing o reduce the spread of Covid-19: Behavioural science and disease prevention guidance for public health*https://www.bps.org.uk/sites/www.bps.org.uk/files/Policy/Policy%20-%20Files/Optimising%20physical%20distancing%20to%20reduce%20the%20spread%20of%20Covid-19.pdf (accessed on 13 August 2021)	19-05-21	02-08-21	6	16	4	7
**Guidance/Thought piece** [44]	**Drury et al., (2021).** The psychology of ‘Freedom Day’: How did the public behave https://thepsychologist.bps.org.uk/psychology-freedom-day (accessed on 13 August 2021)	04-08-21	09-08-21	5	15	1	3

## Data Availability

All guidance documents and reviews detailed in this paper are available through open-access by following the relevant hyperlinks.

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
