# Peer review of "Template for Rapid Iterative Consensus of Experts (TRICE)"

_ijerph, 2021, doi:10.3390/ijerph181910255_

Round 1

Reviewer 1 Report

I highly appreciate the authors’ (and members in the taskforce) contributions (as volunteers without funding) to creating this manuscript during the global pandemic. It shows how a group of health psychologists, practitioners, and policymakers collaborated with each other and created COVID-19 guidance and review documents using consensus methods. Although the manuscript is not a formal review or empirical paper, it provides valuable information to synthesize the knowledge, a collaborative approach to respond to the pandemic, and an implication to the general public beyond local/international health organizations (e.g., CDC, WHO). Thus, the article deserves publication. I wish members of different political parties, health organizations, and academia can use this approach to make their decisions collectively in the U.S. I have a few minor suggestions.  

It seems crucial to know about the 17 members of the task force. Some info was in the main text and table 1. Still, it would be better (and transparent) if the authors can provide an appendix including each member’s (or leader’s) name, affiliation, a brief bio and other credentials. Please provide details on the composition of the team. Were they appointed by the organizations or nominated and then elected by the board? Are all members the authors? It seems two members (17 members but 15 authors) withdrew from the authorship of this document.

Twenty documents were generated. First, were all the documents created after forming the task force? Second, how were these documents created? Some of the documents were “published” on the organizational website, but some were peer-review publications. I am not questioning the validity of the documents, but readers may want to know how the members of the task force contribute their expertise. The authors can provide one specific example to demonstrate the process of creating the guidance document. Third, it ranges from a single meeting to nine meetings to create one guidance document. What were the challenges (topics or the contents) and disagreements? How were the disagreements resolved? The authors should provide an example.  

I wonder if the authors have some data to demonstrate the usage of the guidance/review documents in academia, media, and the general population in the UK beyond the expected implications.  

Reviewer 2 Report

Add related literature regarding the study.

Reviewer 3 Report

This is a much-needed methodology. In response to the Westminster daily briefings, I started a series of short analytical papers outlining deficiencies in the reported deaths in the UK and then around the world. A well-written piece and just a few minor suggested changes.

Line 75 the need for face-to-face meetings has been diminished by the switch to online meetings during the pandemic.

Table 1: Is there any way to left-align the bullet points?

Do you need to make any comments about using the method to inform the public?
